# DPP9: Comprehensive In Silico Analyses of Loss of Function Gene Variants and Associated Gene Expression Signatures in Human Hepatocellular Carcinoma

**DOI:** 10.3390/cancers13071637

**Published:** 2021-04-01

**Authors:** Jiali Carrie Huang, Abdullah Al Emran, Justine Moreno Endaya, Geoffrey W. McCaughan, Mark D. Gorrell, Hui Emma Zhang

**Affiliations:** 1Centenary Institute, Faculty of Medicine and Health, The University of Sydney, Sydney, NSW 2006, Australia; j.huang@centenary.org.au (J.C.H.); a.alemran@centenary.org.au (A.A.E.); jend8597@uni.sydney.edu.au (J.M.E.); g.mccaughan@centenary.org.au (G.W.M.); 2AW Morrow GE & Liver Centre, Royal Prince Alfred Hospital, Camperdown, NSW 2050, Australia

**Keywords:** DPP9, SNPs, hepatocellular carcinoma, survival, TCGA, DPP4 gene family

## Abstract

**Simple Summary:**

This in silico study aimed to investigate associations between dipeptidyl peptidase (DPP) 9 mRNA expression, survival and gene signature in human hepatocellular carcinoma (HCC). *DPP9* loss-of-function exonic variants were mostly associated with cancers. In HCC patients, DPP9 and the closely related genes *DPP4* and *DPP8* were upregulated in liver tumors. High co-expression of genes that were positively correlated with *DPP4*, *DPP8* and *DPP9* was associated with poor survival in HCC patients. These findings strongly implicate the *DPP4* gene family, especially *DPP9*, in the pathogenesis of human HCC and therefore encourages future functional studies.

**Abstract:**

Dipeptidyl peptidase (DPP) 9, DPP8, DPP4 and fibroblast activation protein (FAP) are the four enzymatically active members of the S9b protease family. Associations of DPP9 with human liver cancer, exonic single nucleotide polymorphisms (SNPs) in *DPP9* and loss of function (LoF) variants have not been explored. Human genomic databases, including The Cancer Genome Atlas (TCGA), were interrogated to identify *DPP9* LoF variants and associated cancers. Survival and gene signature analyses were performed on hepatocellular carcinoma (HCC) data. We found that *DPP9* and *DPP8* are intolerant to LoF variants. *DPP9* exonic LoF variants were most often associated with uterine carcinoma and lung carcinoma. All four *DPP4*-like genes were overexpressed in liver tumors and their joint high expression was associated with poor survival in HCC. Increased *DPP9* expression was associated with obesity in HCC patients. High expression of genes that positively correlated with overexpression of *DPP4*, *DPP8*, and *DPP9* were associated with very poor survival in HCC. Enriched pathways analysis of these positively correlated genes featured Toll-like receptor and SUMOylation pathways. This comprehensive data mining suggests that DPP9 is important for survival and that the DPP4 protease family, particularly DPP9, is important in the pathogenesis of human HCC.

## 1. Introduction

Dipeptidyl peptidase (DPP) 9, DPP4, DPP8 and fibroblast activation protein (FAP) are the enzymatic members of the DPP4 family of serine proteases and have been implicated in cancer pathogenesis [1]. DPP9 is ubiquitously expressed in tissues [2] and has diverse roles in cell behaviors [3], immune regulation [4,5,6] and cancer [7,8,9]. DPP9 can interact with H-Ras, which is a key molecule of the epidermal growth factor receptor and PI3K/Akt signaling pathways; these pathways are important for cell survival, proliferation and apoptosis [3]. DPP9 inhibition stimulates the immune system by activating pyroptosis in monocytes and macrophages, in which inhibition of DPP9 causes formation of multiprotein complexes called inflammasomes, which activates caspase-1 and generates pyroptosis [3]. A study using a non-small-cell-lung cancer (NSCLC) model demonstrated that knockdown of DPP9 could inhibit lung cancer cell proliferation, migration and tumorigenesis. Overexpression of DPP9 in NSCLC is independently associated with poor 5-year overall survival [7]. Similarly, in colorectal cancer, greater DPP9 expression is associated with poor prognosis [8]. In contrast, in patients with oral squamous cell carcinoma, lower DPP9 expression correlates with poor survival [9]. These data suggest that DPP9 has different roles in various types of cancers.

Liver is rich in DPP9 enzyme activity [2], and DPP9 mRNA is upregulated in inflamed and fibrotic mouse liver [10]. Downregulating DPP9 or blocking its enzyme activity reduces cell adhesion and migration in the Huh7 human hepatoma cell line [11]. The role of DPP9 in human liver cancer has not been studied, particularly in liver hepatocellular carcinoma (HCC). HCC is the most common primary liver cancer [12]. The major risk for HCC is chronic liver injury, with causes including non-alcoholic fatty liver disease (NAFLD), obesity, smoking tobacco, chronic hepatitis B or C, and chronic excessive alcohol consumption [13,14,15]. NAFLD, obesity and associated metabolic diseases are becoming the leading cause of HCC [16].

The DPP4 enzyme family is the S9b subfamily of the prolyl oligopeptidase (S9) family and share 27–60% protein sequence identity with each other [17,18]. The proteases in the DPP4 enzyme family are able to hydrolyze peptides after a proline residue and can remove a dipeptide from the N-terminus of substrates [18,19]. Very few proteases have this capability. DPP4 and FAP are expressed on the cell surface, whereas DPP8 and DPP9 are intracellular [1]. DPP8 and DPP9 share the greatest similarity in protein sequence and crystal structure in this gene family [20]. There are two isoforms of DPP9 with distinct subcellular localizations: the long form (892 amino acids) is in both cytosol and nucleus because it contains a nuclear localization sequence while the short form (863 amino acids) is only cytosolic [21]. The S9b proteases are active as dimers [22], with each monomer having two domains, the α/β-hydrolase domain and the β-propeller domain [20]. The β−propeller domain binds to protein ligands, and DPP9 has been found to bind to SUMO-1 [23] and filamin A [6] and associate with H-Ras [3], which are probably involved in roles of DPP9 in cell adhesion and migration. Selective inhibitors for DPP9 versus DPP8 are lacking, so individual functions of DPP9 are poorly resolved and derive primarily from genetic manipulations of mice and cell lines [1,24,25].

Large-scale human genetic databases have become increasingly powerful resources to study the impacts of somatic mutation in diseases, especially in cancer. Human gene sequence variants that are predicted to cause loss of function (LoF) of an encoded protein provide valuable data on studying human gene inactivation [26,27]. Genome-wide association studies (GWAS) have found several risk loci mapping to the DPP9 gene on chromosome 19 (19p13.3) that contribute to human diseases, including alcoholism [28], interstitial lung abnormalities [29], pulmonary fibrosis [29,30] and adolescent idiopathic scoliosis [31]. The intronic single nucleotide polymorphism (SNP) rs113683471 in *DPP9* is associated with alcoholism [28]. Intronic SNPs rs12610495 and rs2109069 in *DPP9* are genome-wide significant loci for severe COVID-19, idiopathic pulmonary fibrosis and idiopathic interstitial pneumonia [29,30,32,33]. However, there are limited information on exonic SNPs in *DPP9* and *DPP9* LoF variants, or their associations with diseases.

This study identifies *DPP9* exonic LoF variants in human genomic databases and their associations with cancers. This study also reports the first comprehensive data mining of associations between *DPP9* mRNA expression, survival and gene signature in human HCC.

## 2. Materials and Methods

### 2.1. Protein Sequences

The amino acid sequences of the short (863 amino acids) and long forms (892 amino acids) of DPP9 were downloaded from the UniProt Consortium (https://www.uniprot.org/uniprot/Q86TI2, accessed on 7 April 2020). The long form of DPP9 and its amino acid numbering was used in this study. Protein crystal structure of the four enzymatic members of the DPP4 enzyme family were depicted using the PyMOL Molecular Graphics System Version 2.4.2, and a multiple-sequence alignment was made using a Clustal Omega bioinformatics tools for multiple sequence alignment (https://www.ebi.ac.uk/Tools/msa/, accessed on 6 March 2021) (Appendix A).

### 2.2. Public Databases for Accessing Loss of Function Variants

Public databases assessed in this study are listed in Table 1.

### 2.3. Loss of Function Variants

The exonic variants that are predicted to cause LoF of *DPP9* were exported from gnomAD, TCGA and COSMIC. Only the LoF variants that are annotated as either stop gained or frameshift in the databases were studied. The numbering of the amino acid sequence of DPP9 in GnomAD is of the short form of DPP9. Therefore, to use the DPP9 long form, 29 was added to all protein sequence numbers for the gnomAD *DPP9* LoF analyses. *DPP9* variants from TCGA and COSMIC were found to be of the long form. Variants with a sequence number mismatch with the long form of DPP9 were excluded. In addition, if frameshift variants exported from gnomAD and TCGA occurred where there is no corresponding structural residue, these frameshift variants were considered unlikely to be *DPP9* exonic LoF variants, and thus were excluded from this study.

### 2.4. Intronic Variants

The two *DPP9* intronic variants rs12610495 and rs2109069 have the smallest *p* values that have been associated with critical illness caused by COVID-19 [33]. More genetic information about these two intronic variants was exported from PhenoScanner V2 [40].

### 2.5. Genetic Variant Tolerance Parameters

The observed to expected ratio (o:e) of LoF, the probability of LoF intolerance (pLI) and haploinsufficiency score were retrieved from gnomAD and DECIPHER [38]. Tolerance percentile and GDI values were retrieved from GeneCards human gene database.

### 2.6. Gene Correlation Analysis

UALCAN is a comprehensive web database for investigating complete genetic or molecular data of cancers and allows in silico validation of genes of interest [41]. It is available online at http://ualcan.path.uab.edu/index.html, accessed on 16 October 2020. Both positively and negatively correlated genes of *DPPs* and *FAP* were accessed using liver hepatocellular carcinoma (LIHC; HCC) dataset of TCGA. The expression level of each gene was displayed as transcripts per million (TPM).

### 2.7. Survival Analysis

OncoLnc database contains TCGA survival data to mRNA expression level for 8647 patients from 21 cancer projects [42]. Survival data for 360 patients with HCC and 540 patients with UCEC were exported from OncoLnc. Normal patients’ data (*n* = 50) were exported from UALCAN [41]. HCC patient survival was also analyzed based on patients’ BMI (Table 2). High and low levels of gene expression were stratified based on either median expression, or the upper or lower quartile, based on the biological questions we were addressing.

SurvExpress is an online database for cancer gene expression data using survival analysis, which is able to access the influence of combined high expression of multiple genes on survival in a specific cancer type [43]. The genes that correlate in-common amongst the DPP4 gene family were imported into SurvExpress and analyzed for their associations with HCC patient survival (June 2016 dataset). Patient survival time and status were exported from SurvExpress and Kaplan-Meier curves were plotted.

### 2.8. Reactome Pathways Enrichment Analysis

Reactome enrichment pathways of positively or negatively correlated genes in-common between *DPP9* and *DPP8*, and of the positively correlated genes in-common amongst *DPP9, DPP8* and *DPP4* were explored using ConcensusPathDB (CPDB) [44]. Genes were imported into ConcensusPathDB and identified as gene symbols. The analysis was performed under gene set analysis and over-representation analysis. In ConsensusPathDB, each *p* value was calculated in a hypergeometric test based on the number of entities present in both the predefined set and a user defined list of entities, and the *p* values were corrected for multiple testing using false discovery rate method.

### 2.9. Data and Statistical Analyses

Figures and Kaplan-Meier curves were generated in GraphPad Prism (version 8.4.2, San Diego, CA, USA). Data statistics were analyzed in GraphPad Prism. Survival analysis was performed in Prism using logrank (Mantel-Cox) to assess the significance between the groups. The Mann-Whitney U test was used to compare differences between two independent groups. The Kruskal-Wallis one-way analysis of variance with Tukey’s post-hoc test was used to compare differences amongst multiple groups. Significance was assigned to *p* values < 0.05, <0.01 or <0.0001.

## 3. Results

### 3.1. DPP9 Loss of Function (LoF) Variants and Disease Associations

Large-scale human genetic databases provide useful tools to study protein LoF variants and their associations with diseases. *DPP9* LoF has not been explored previously. Therefore, we assessed data from several human genomic databases to identify *DPP9* LoF and potentially associated diseases. Gaining a premature termination codon (PTC), or stop codon, anywhere in a gene of the DPP4 gene family is expected to ablate enzyme activity [45]. We have shown previously that mature termination mutants of *DPP8* are inactive, even if lacking only the C-terminal alpha-helix [45]. Similarly, any frameshift variant is almost certain to lack activity due to nonsense sequence causing a PTC. Therefore, the exonic gene variants considered to be LoF are very likely true LoF.

In gnomAD, there were 18 exonic variants of *DPP9* that were tagged as LoF. However, examining the sequence of long-form DPP9 in the UniProt Consortium database revealed that some variants were mismatched with UniProt, and thus were excluded. Only exonic LoF variants that were annotated as either stop-gained or frameshift variants were included in this study. We identified three frameshift variants and seven stop gained variants that were exonic in gnomAD (Table 3). There was no record of associated diseases for each mutation variant. In TCGA, we identified eight *DPP9* LoF variants (Table 4), and none of these variants were the same as found in gnomAD. Four *DPP9* variants were designated as stop gained, with three of them associated with primary uterine corpus endometrial carcinoma (UCEC). Another four *DPP9* variants were designated as frameshift mutation, with half of them found in patients diagnosed with stomach adenocarcinoma (Table 4). None of the *DPP9* variants in TCGA were found to be associated with HCC. There were many more *DPP8* than *DPP9* LoF gene variants in TCGA, found in diverse types of cancer, mostly in UCEC and one in HCC (Appendix A). In a separate database, COSMIC, eleven *DPP9* variants had been tagged as nonsense mutation and were substitution mutations resulting in PTC codons (Table 5). These variants were associated with different types of carcinoma, most often lung carcinoma. All variants that were likely to cause LoF for *DPP8* or *DPP9* have a PTC. All exonic *DPP9* variants found in each of the three databases were mapped to the DPP9 protein structure (Appendix A).

There were 22 *DPP8* nonsense mutations in COSMIC, most often in colon adenocarcinoma and one in HCC (Appendix A). *DPP9* variants Glutamic acid (Glu) 154 to PTC and Arginine (Arg) 303 to PTC were both reported in both TCGA and COSMIC, but with different nucleotide substitutions. Another anomaly was that in COSMIC, both endometrioid carcinoma and colon adenocarcinoma were associated with the Arg303* variant, but in TCGA only endometrioid carcinoma was recorded.

### 3.2. Intronic SNPs in DPP9

Intronic SNPs rs12610495 and rs2109069 in DPP9 are genome-wide significant loci for severe COVID-19, idiopathic pulmonary fibrosis and idiopathic interstitial pneumonia [29,30,32,33]. These SNPs are located 5′ to the open reading frame, at nucleotides 4717672 and 4719443 of chromosome 19. They are important DPP9 SNPs because they are relatively common, with allelic frequencies of 0.706 to 0.828 for rs12610495 and 0.186 to 0.321 for rs2109069 (Appendix A). These SNPs have been associated with lower DPP9 expression levels in human lung [33]. We found that these SNPs are not associated with liver fibrosis or cancer in the PhenoScanner database.

### 3.3. Genetic Variation Tolerance in DPP9 and the DPP4 Gene Family

Some databases incorporate scoring systems that determine tolerance towards genetic variations. We analyzed genetic variation tolerance in the DPP4 gene family using five scores: observed to expected ratio (o:e), probability of LoF intolerance (pLI), haploinsufficiency score, tolerance percentile, and gene damage index (GDI) scores. When o:e is approximately 0.1, the gene is haploinsufficient and heterozygous LoF is very probably not tolerated; when o:e is between 0.1 and 0.5, the heterozygous LoF may be tolerated; when o:e is close to 1, LoF is tolerated [46]. DPP8 and DPP9 had low o:e, little more than 0.1, DPP4 had a moderate o:e less than 0.5 and FAP has a high o:e (Table 6). The probability of LoF intolerance (pLI) is the probability of a given gene being haploinsufficient; genes with pLI ≥ 0.9 are extremely intolerant to LoF variation and genes with pLI ≤ 0.1 are LoF tolerant [46]. The pLI scores for DPP9, DPP8, DPP4 and FAP (Table 6) suggested that DPP8 and DPP9 LoF variants are poorly tolerated. Haploinsufficient genes only contain a single copy of a diploid genome that changes the organism phenotype from a normal to disease state, and many of the LoF-associated haploinsufficient genes contribute to metabolic disorders and tumorigenesis [47]. The haploinsufficiency score was greater for DPP9 than for DPP8, DPP4 or FAP, and, like the o:e and pLI, indicated that DPP9 is likely to exhibit haploinsufficiency whereas DPP4 and FAP are unlikely to exhibit haploinsufficiency. The gene damage index (GDI) scores for these four enzymes were low (Table 6), suggesting that LoF variants are likely to be disease-causing [48]. Genes in the 25th tolerance percentile and below are considered relatively intolerant to variation. Unlike the other haploinsufficiency indications, GDI and tolerance percentile did not discriminate amongst the four genes (Table 6).

### 3.4. Association of DPP9 Expression with Patient Survival in HCC and UCEC

Previous studies reported associations of DPP9 expression with patient survival in non-small-cell lung cancer [7], colorectal cancer [8] and oral squamous cell carcinoma [9]. Many DPP8 and DPP9 variants were identified in patients with UCEC and HCC in our study (Table 4 and Appendix A). Therefore, we analyzed whether DPPs expression is associated with patient survival in HCC and UCEC. Clinical parameters and mRNA expression of DPPs in patients with HCC and UCEC were exported from TCGA. Patient demographics are presented in Appendix A. Patients were stratified by upper or lower quartile expression of DPPs. Kaplan-Meier analysis for DPP9, DPP8, DPP4 and FAP mRNA suggested that expression of none of these four genes was associated with HCC overall survival (OS) (Figure 1A–D). In contrast, patients with low DPP9 mRNA expression had significantly shorter survival time for UCEC (Figure 1E). Less than 15% of UCEC patients with low DPP9 expression had survived for 10 years after diagnosis. Unlike DPP9, DPP8, DPP4 and FAP expression did not associate with OS in UCEC (Figure 1F–H). This data indicates that DPP9 might have an important role in UCEC.

### 3.5. Association of DPP9 Expression with Clinical Parameters and Survival Outcome in HCC

We next explored closely the association of DPP9 with clinical parameters relevant to HCC, including gender, age, BMI and cancer stages. In HCC patients, all four DPP genes were significantly upregulated in tumors compared to normal tissue (Figure 2A–D). Epidemiology studies have shown that globally, the numbers of male HCC patients are two to four times greater than female HCC patients [15,49]. Our results showed that significantly greater expression of DPP9 was detected in liver tumors from females than males (Figure 2E). It may be expected that HCC patients with greater age and advance stage of cancer will have more expression of DPP9 and therefore poorer prognosis. However, DPP9 expression did not differ significantly amongst six age groups (Figure 2F), or amongst different cancer stages (Figure 2G). In addition, we used Cox proportional hazards model with gender as a covariate to re-evaluate the associations between DPP9 expression and survival. DPP9 expression was not significantly associated with poor prognosis based on patients’ gender (Appendix A).

Obesity is a well-known risk factor for HCC [16], so we examined DPP9 expression with HCC patient survival based on body mass index (BMI; BMI categories in Table 2). We stratified HCC patients based on BMI and found that DPP9 mRNA was significantly upregulated in the obese and extreme obese patients compared to patients with normal body weight (Figure 3A). Survival analysis suggested that DPP9 was not significantly associated with patient survival when patients are overweight (Figure 3B), or obese/extremely obese (Figure 3C). We then used a Cox proportional hazards model with BMI as a covariate to evaluate associations between DPP9 expression and survival. DPP9 expression was not significantly associated with poor prognosis based on patients’ BMI (Appendix A). High DPP9 expression in the overweight group had a hazard ratio of 1.54, but it lacked statistical significance.

### 3.6. Association of Expression of DPPs with Poor Survival of Patients with HCC

To further investigate DPP4 gene family expression in liver cancer prognosis, we stratified DPP9, DPP8 and DPP4 based on median expression in HCC. High co-expression of DPP9, DPP8 and DPP4 was not associated with OS (Figure 4A). However, high co-expression of DPP9, DPP8, DPP4 and FAP in the liver tumor was significantly associated with poor survival in HCC patients (Figure 4B). This data strengthens evidence that FAP might contribute to poor outcomes, possibly due to its role in liver cirrhosis and steatosis, which is strongly associated with poor liver function and liver cancer [50,51].

### 3.7. Genes Correlated with DPPs Significantly Associated with Poor Prognosis in HCC and Multiple Oncogenic and Epigenetic Pathways

Having shown that DPP9, DPP8, DPP4 and FAP were overexpressed in liver tumors (Figure 2A–D), and that high co-expression of the DPP4 gene family members was significantly associated with shorter survival times in HCC (Figure 4B), we next assessed the gene expression signature correlated with the DPP4 gene family in HCC patients to gain mechanistic insights. Firstly, we analyzed the two homologs DPP8 and DPP9 and found that 661 genes were positively correlated in-common between DPP8 and DPP9 (Figure 5A). Reactome pathway enrichment analysis revealed that these genes were associated with tumor suppressor TP53 regulation and with chromatin remodeling pathways in HCC (Figure 5B). In contrast, only 10 negatively correlated genes in-common between DPP8 and DPP9 were identified and they were associated with metabolism related pathways (Appendix A).

A similar approach was undertaken to study the gene expression signature correlated with all three genes, DPP9, DPP8 and DPP4. We found that 268 genes were positively correlated and in-common amongst DPP9, DPP8 and DPP4 in HCC patients (Figure 6A), and stratified the expression of these genes based on median expression. Most importantly, high expression of these positively correlated gene signatures was significantly associated with poor prognosis in HCC patients (Figure 6B, logrank *p* < 0.0001).

To gain a better understanding of how DPPs may influence survival, we analyzed the biological pathways that were enriched in genes that were positively correlated with DPPs. There were 46 significantly enriched pathways according to reactome annotations for those 268 genes (Appendix A). Amongst the 15 top-ranked pathways (Figure 6C), the Toll-like receptors (TLRs) cascade, especially the TLR9 and TLR7/8 cascade, MyD88 dependent cascade initiated on endosome, N-glycosylation and SUMOylation pathways were dominant. When FAP was added to this analysis, only seven in-common genes emerged, most of which are the zinc finger gene regulators LIMS1, PHF20L1, ZEB1 and ZNFX1 (Appendix A).

## 4. Discussion

This is the first set of analysis of human genomic databases to examine *DPP9* exonic LoF variants and the of association of DPP9 expression with cancers. There were 27 unique *DPP9* LoF variants found from our analyses of gnomAD, TCGA and COSMIC. Notably, patients with *DPP9* LoF variants were most commonly diagnosed with cancers. The two important *DPP9* intronic variants (rs12610495 and rs2109069) were found to be associated with pulmonary fibrosis and severe COVID, as shown previously [29,30,32,33], but not with liver cancer. Uterine corpus endometrial carcinoma (UCEC) and lung carcinoma were the most common diagnosed cancers in patients with *DPP9* LoF variants. It is likely that *DPP9* LoF variants will result in lowered abundance of DPP9 enzyme activity. Our datamining evidence that *DPP9* LoF was associated with lung carcinoma perhaps does not align with a previous study that linked high *DPP9* mRNA expression with poor survival in human lung cancer [7]. Perhaps the role of DPP9 that is dominant differs in different cancer types and differs depending upon whether high or low DPP9 expression occurred in the tumor or was life-long. Due to the limited case numbers of *DPP9* LoF variants, we were unable to associate these variants with patient survival.

We observed that *DPP8* and *DPP9* LoF variants were more commonly diagnosed with a gynecological cancer, such as UCEC, cervical cancer or ovarian cancer. Together with the observation that low DPP9 expression was associated with poor survival in UCEC (Figure 1E), it might suggest that *DPP9* is a suppressor of gynecological cancer, which would align with the recently discovered role of DPP9 in BRCA2 homeostasis and DNA repair [52]. However, in both TCGA and COSMIC, there was limited clinical data concerning patient outcomes, especially in COSMIC, where many cases were filed with unknown gender, age and diseases. This is a common limitation of in silico investigations.

Our investigation found that *DPP9* and *DPP8* are LoF intolerant in humans. Genetic variation tolerance parameters universally suggested that *DPP9* was LoF intolerant (Table 6). LoF intolerance has been associated with successful targets of medicinal therapies for cancer [26]. The necessity of DPP9 is conserved in mice, where constitutive homozygous ablation of DPP9 enzyme activity is neonate lethal [25,53,54,55]. There were fewer than expected numbers of LoF variants observed in the three large-scale databases that we interrogated. The reason could be that the association with gynecological cancers, mentioned above, also associates with poor reproductive fitness. In addition, people who are homozygous or biallelic for a *DPP9* LoF variant may not survive to be included in the searched databases, or survivors possibly possess a modification in a molecular pathway that compensates for DPP9 deficiency. An example is that deletion of NLRP1 and its downstream signaling molecules rescues *DPP9* deficient mice from lethality [56]. Another consideration is that the LoF *DPP9* variants in genomic databases are expected to be heterozygous and would therefore have lifelong low *DPP9* expression throughout the body. Taken together, these data show that *DPP9* is LoF intolerant but suggest that *DPP9* LoF variants do not necessarily cause early death.

The databases contained some data mismatches about *DPP9* sequences, presumably due to automation of data importation. There were more LoF variants in gnomAD than in COSMIC, but some of the listed LoF locations were not found in either long form or short form DPP9. Moreover, in both TCGA and COSMIC, variants of Arg303* and Glu154* were reported, but the mutations of nucleotides differed between the two databases. Also, in patients with the Arg303* variant, COSMIC reported both endometrial carcinoma and large intestinal adenocarcinoma, while TCGA only reported endometrial carcinoma. Hence, using a single database can be misleading and unmatched information is inevitable when using multiple databases. Therefore, database-derived information needs to be carefully and critically assessed.

Our TCGA datamining on the DPP4 gene family showed that in HCC patients, *DPP9, DPP8, DPP4* and *FAP* were all overexpressed in liver tumors. Most notably, high expression of all four genes showed poorer survival in HCC patients (Figure 4B). Therefore, DPP targeted therapy might need to target all four enzymes together. Supporting this concept, VbP (Talabostat; PT-100; BXCL-701), which exerts potent inhibition of DPP9 and DPP8 and moderate inhibition of DPP4 and FAP, has been shown to lessen tumor burden in lung cancer and sarcoma models [57,58,59].

We have examined the associations between *DPP9* mRNA expression and several potential prognostic factors, including gender, age, cancer stage and BMI. The impacts of these factors on HCC prognosis remain controversial. Our data analysis showed that *DPP9* expression is greater in females than males. Even though age is known not to be an independent risk factor associated with cancer survival, the cumulative 10-year survival rate was significantly higher in younger patients [60]. Here, we saw no significant difference in *DPP9* expression among different age groups. Moreover, patients diagnosed with advanced cancer stages would generally have shorter life spans, but our results showed that *DPP9* expression was not associated with cancer stage. Taken together, the difference in survival probability amongst age groups and patients at different cancer stages may not be attributed to the expression of *DPP9*. Obesity is an independent risk factor for developing HCC. Our data showed that *DPP9* expression was significantly upregulated in HCC patients who were obese/extremely obese, but significantly downregulated in overweight HCC patients. However, survival analysis on BMI showed no significant association with *DPP9* expression, despite a hazard ratio greater than 1. As liver cancer is a multifactorial disease, further investigations may combine several risk factors together and investigate multiple comparisons between these parameters.

Enriched biochemical pathways analysis provided insights into differentially expressed genes correlated with *DPPs* in human HCC. Positively correlated genes in-common between *DPP8* and *DPP9* were associated with oncogenic and chromatin remodeling pathways in HCC, whilst 10 negatively correlated genes were associated with metabolism related pathways (Appendix A). Mutation of the tumor suppressor *TP53* is associated with poor outcome for HCC patients [61]. The top ranking *TP53* regulation pathway in our analyses indicates that *DPP8* and *DPP9* overexpression could be oncogenic in human HCC. The downregulation of metabolism genes with *DPP8* and *DPP9* upregulation concords with previous data showing that DPP9 deficiency dysregulates metabolic pathways in neonatal liver and gut [54], and that DPP9 hydrolyses metabolism related substrates including adenylate kinase 2 [62,63] and nucleobindin1 [64] and regulates preadipocyte differentiation [65]. Moreover, people with obesity are more likely to develop HCC [66,67]. *DPP9* expression was significantly upregulated in HCC patients who were obese and extremely obese compared to those with normal weight. Taken together, this information suggests that DPP9 has an important role in liver manifestations of the metabolic syndrome, which drives progression to NASH, fibrosis and HCC.

The high expression of positively correlated gene signatures in-common amongst DPPs were associated with considerably shorter survival time in HCC patients. Amongst the identified reactome pathways, the TLR cascade, glycosylation and SUMOylation were the most frequent pathways enriched with increased *DPP9* mRNA expression. No association between DPP9 and glycosylation is known. SUMOylation is mostly associated with SUMOylation of chromatin organization proteins, and SUMOylation of DNA damage response and repair proteins, which could have roles in cancer development. SUMO1 is a DPP9 interacting protein, acting as an allosteric regulator of DPP9 and thereby influencing its enzymatic activity [23]. TLR7/8 and TLR9 signaling pathways have not been linked to DPP9 previously. The expression of TLRs is low in healthy liver [68]. TLRs are upregulated in HCC tissues [69], and inhibition of TLR7 and TLR9 eliminates the proliferation of liver cancer cells [69,70]. Therefore, overexpression of *DPPs* might activate these TLR cascades, thereby inducing overactivation of the innate immune system and pathological changes.

## 5. Conclusions

This study reports the first comprehensive data mining of associations between *DPP9* expression, survival and gene expression signatures in human HCC. *DPP9* was found to be very LoF-intolerant and its LoF exonic variants were mostly associated with cancers. Two intronic variants that lower *DPP9* expression and have been linked to lung fibrosis and COVID-19 were not associated with HCC or liver fibrosis. In HCC patients, *DPP9* and its closely related genes were significantly upregulated in liver tumors. High co-expression of genes that were positively correlated and in-common amongst *DPP4, DPP8* and *DPP9* were associated with poor survival of HCC patients, possibly via affecting the following enriched pathways: Toll-like receptors (TLRs), glycosylation and SUMOylation. These findings strongly implicate the DPP4 gene family, especially DPP9, in the pathogenesis of human HCC and therefore requires future functional studies.

## Figures and Tables

**Figure 1 cancers-13-01637-f001:**
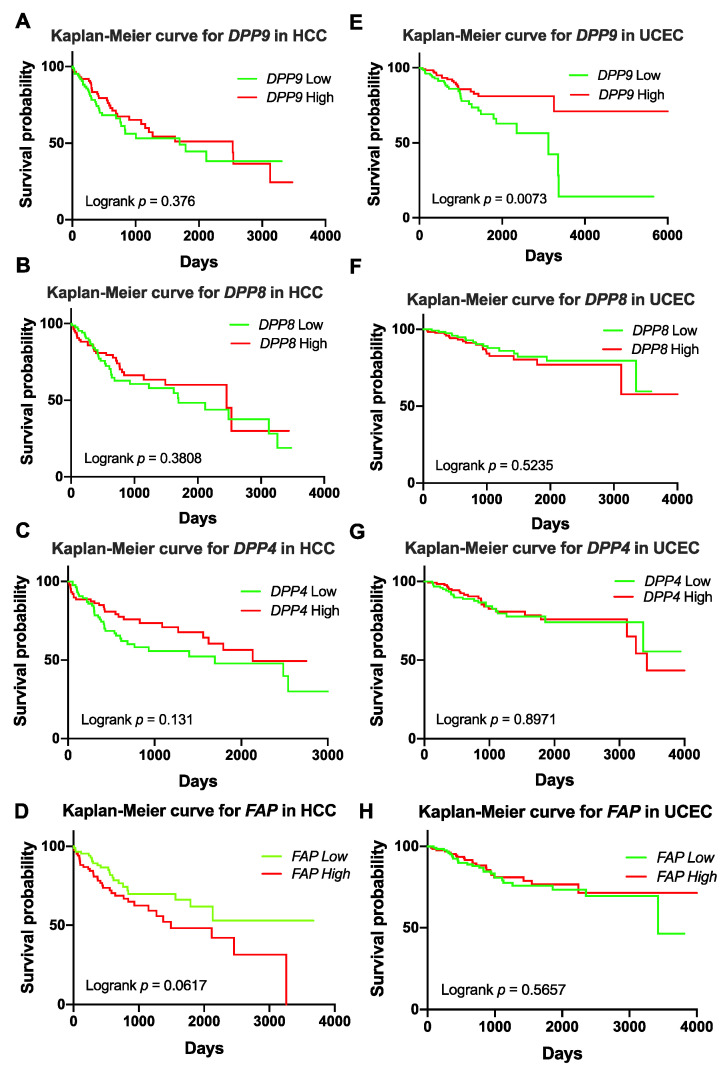
Kaplan-Meier curves for *DPP9, DPP8, DPP4* and *FAP* in liver tumors (hepatocellular carcinoma, HCC; (**A**–**D**)) (*n* = 360) and uterine corpus endometrial carcinoma (UCEC; (**E**–**H**)) (*n* = 540). High or low was defined as the mRNA expression values in the upper or lower quartile, respectively. *p* values were calculated by logrank (Mantel-Cox) test.

**Figure 2 cancers-13-01637-f002:**
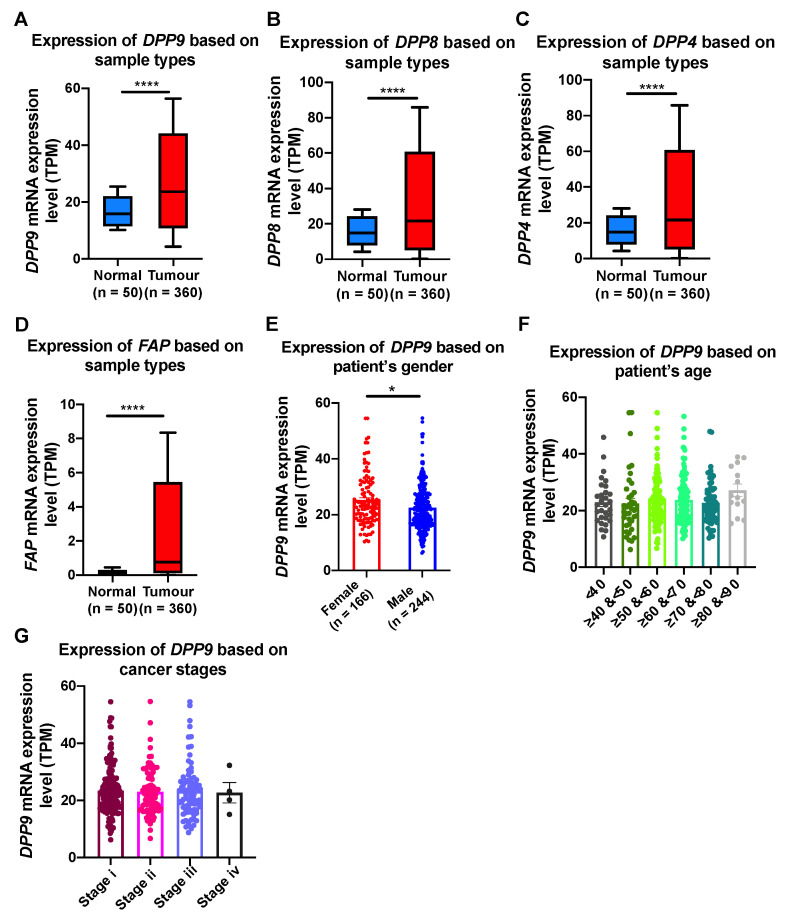
The relative mRNA expression level of *DPP9, DPP8, DPP4* and *FAP* in liver tumors (**A**–**D**) and association of *DPP9* expression with gender (**E**), age (**F**) and cancer stages (**G**) in HCC patients. *p* values were calculated by Mann-Whitney U test (**A**–**E**) and Kruskal-Wallis one-way analysis of variance with Tukey’s post-hoc test (**F**,**G**). * *p* < 0.05, **** *p* < 0.0001.

**Figure 3 cancers-13-01637-f003:**
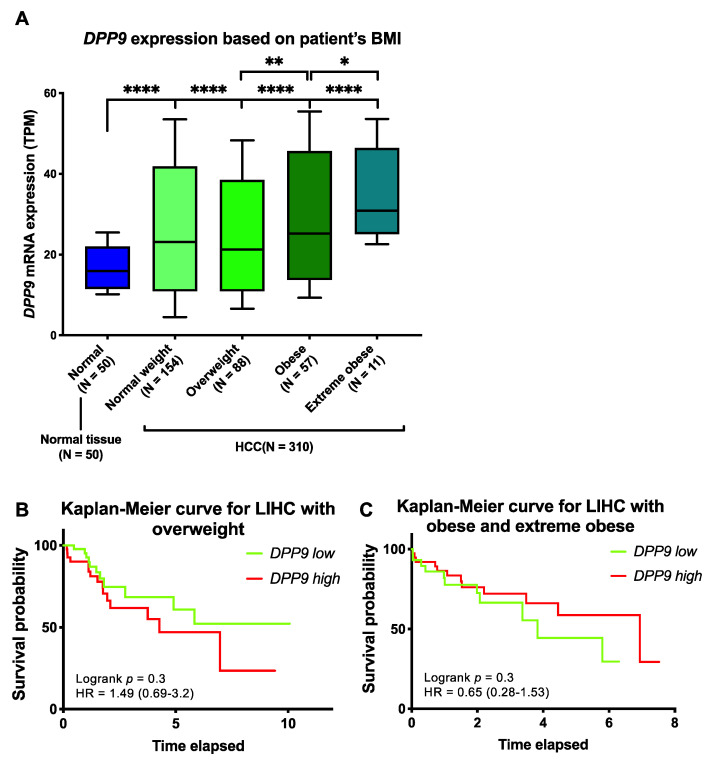
The relative mRNA expression and survival analysis of *DPP9* in HCC based on patient BMI. (**A**) The mRNA expression of *DPP9* based on HCC patient (*n* = 360) BMI compared to non-diseased subjects. *p* values were calculated by Kruskal-Wallis one-way analysis of variance with Tukey’s post-hoc test, with * *p* < 0.05, ** *p* < 0.01, **** *p* < 0.0001. (**B**) Kaplan-Meier curve for *DPP9* expression in overweight HCC patients. (**C**) Kaplan-Meier curve for *DPP9* expression in obese and extreme obese HCC patients. The high (red) and low (green) mRNA expression levels of genes in liver tumors were stratified based on median expression value. *p* values and hazard ratio (HR) with confidence interval were calculated by logrank (Mantel-Cox) test.

**Figure 4 cancers-13-01637-f004:**
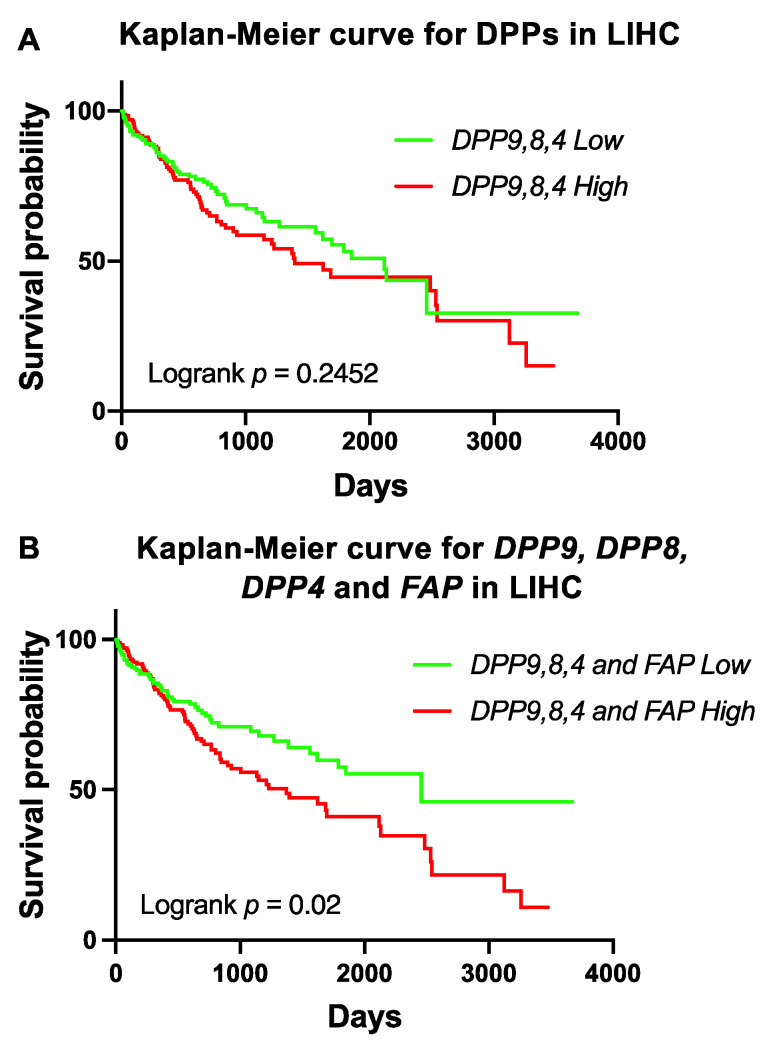
Survival analysis of high co-expression of *DPP9, DPP8, DPP4* and *FAP* in HCC patients. **(A**) Kaplan-Meier curve for high co-expression of *DPP9, DPP8* and *DPP4* in HCC patients (*n* = 360). (**B**) Kaplan-Meier curve for high co-expression of *DPP9, DPP8, DPP4* and *FAP* in HCC patients. The high (red) and low (green) mRNA expression levels of genes in liver tumors were stratified based on median expression value. *p* values were calculated by logrank (Mantel-Cox) test.

**Figure 5 cancers-13-01637-f005:**
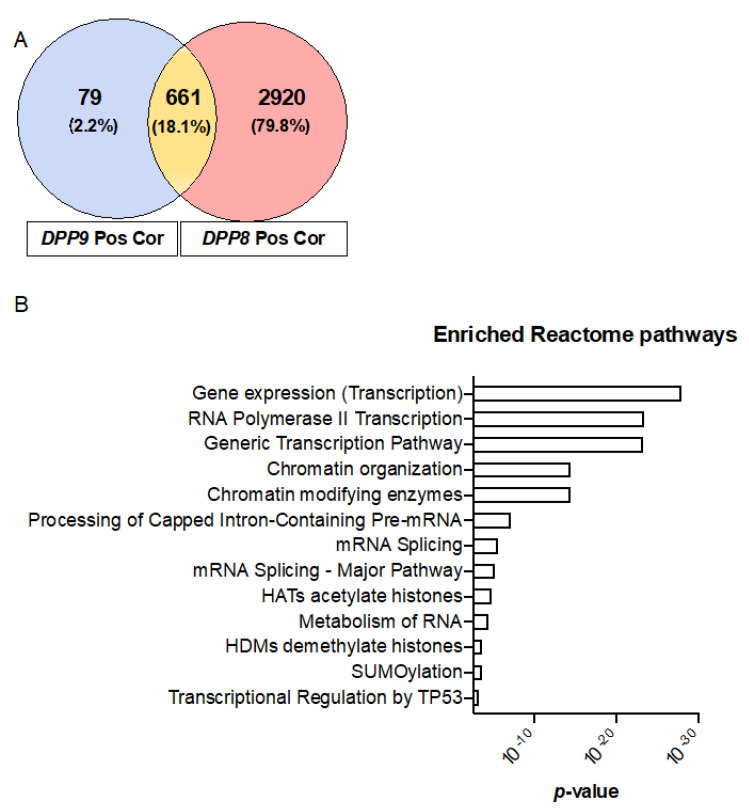
Enriched reactome pathways of genes that were positively correlated with *DPP9* and *DPP8*. (**A**) Venn diagram of the genes that were positively correlated with *DPP9* and *DPP8*, with Pearson correlation coefficient greater or equal 0.5. “Pos Cor” refers to positive correlation. (**B**) Enriched reactome pathways associated with positively correlated genes in-common between *DPP9* and *DPP8*.

**Figure 6 cancers-13-01637-f006:**
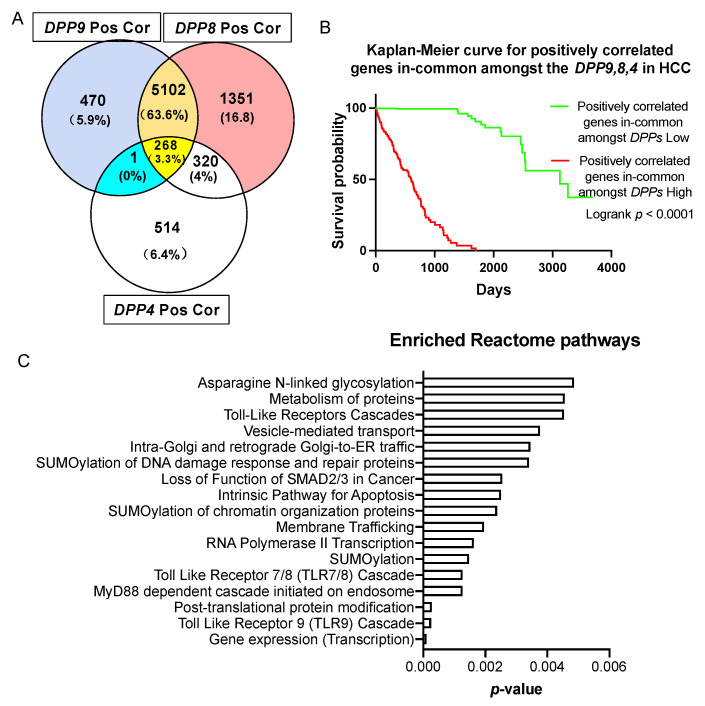
Survival analysis and enriched reactome pathways analysis on genes that were positively correlated with *DPP9, DPP8* and *DPP4*. (**A**) Venn diagram of the gens that were positively correlated with *DPP9, DPP8* and *DPP4*. The gene in the blue section is *TPRA1*. “Pos Cor” refers to positive correlation. (**B**) Kaplan-Meier curve for the 268 genes that were positively correlated and in-common amongst *DPP9, DPP8* and *DPP4* in HCC patients (*n* = 360). The high (red) and low (green) mRNA expression levels of genes in liver tumors were stratified based on median expression value. *p* values were calculated by logrank (Mantel-Cox) test. (**C**) Enriched reactome pathways of genes that were positively correlated in-common amongst *DPP9, DPP8* and *DPP4*.

**Table 1 cancers-13-01637-t001:** Public databases sourced for loss of function data in this study.

Databases	URL	Data Obtained Date	Citation
The Genome Aggregation database (gnomAD)	https://gnomad.broadinstitute.org/	5 April 2020	[34]
The Cancer Genome Atlas (TCGA)	https://portal.gdc.cancer.gov/	20 April 2020	[35]
Catalogue of Somatic Mutation in Cancer (COSMIC)	https://cancer.sanger.ac.uk/cosmic	15 June 2020	[36]
UniProt Consortium	https://www.uniprot.org/	7 April 2020	[37]
DatabasE of genomiC varIation and Phenotype in Humans using Ensembl Resources (DECIPHER)	https://decipher.sanger.ac.uk/	10 May 2020	[38]
GeneCards human gene database	https://www.genecards.org/	12 May 2020	[39]

**Table 2 cancers-13-01637-t002:** Categories of BMI.

Status	BMI Range
Normal weight	18.5–25
Overweight	25–30
Obese	30–40
Extreme obese	>40

**Table 3 cancers-13-01637-t003:** *DPP9* LoF variants from gnomAD. ins = insertion. * Premature termination codon (PTC).

Position	rsID	Reference	Alternate	Protein Consequence	Annotation	Allele Frequency
4682801	rs766329416	CCT	C	p.Gly794Val * 3	frameshift variant	7.89 × 10^−6^
4694820	rs1360075526	AGAAG	A	p.Phe456Ile * 34	frameshift variant	4.02 × 10^−6^
4714106	rs1162979581	GC	G	p.Arg100Pro * 58	frameshift variant	4.1 × 10^−6^
4683566	rs1199678192	G	A	p.Arg752 *	Stop gained	4.02 × 10^−6^
4684713	rs1165785000	G	A	p.Arg714 *	Stop gained	4.1 × 10^−6^
4676614	rs1479652400	C	A	p.Glu881 *	Stop gained	1.25 × 10^−5^
4694808	rs1365336720	G	A	p.Gln461 * ^1^	Stop gained	4.02 × 10^−6^
4694825	rs1418069794	A	ATCTTCTATC	p.Asp454Ile455insArg * Lys	Stop gained	4.02 × 10^−6^
4702672	rs1466943995	G	A	p.Gln276 *	Stop gained	7.53 × 10^−6^
4705965	rs779854175	G	A	p.Arg111 * ^1^	Stop gained	1.2 × 10^−5^

^1^ Gln461 and Arg111: no corresponding residue in the primary structure of DPP9.

**Table 4 cancers-13-01637-t004:** *DPP9* LoF variants in TCGA. Simple somatic mutation (SSM) affected frequency was calculated as the number of cases affected by a specific mutation in a TCGA disease project divided by the number of cases tested for SSM in that disease project in TCGA. ins = insertion. del = deletion. * Premature termination codon (PTC).

Position	Reference	Alternate	Protein Consequence	Annotation	Disease	Number of SSM Affected Cases and Frequency
4685761_4685762insG			p.Asp633Gly * 51	Frameshift	Stomach Adenocarcinoma	1/440 (0.23%)
4684670delT			p.Asn724Thr * 27	Frameshift	Stomach Adenocarcinoma	1/440 (0.23%)
4679893delA			p.Phe843Ser * 10	Frameshift	Colon Adenocarcinoma	1/400 (0.25%)
4676645_4676646insT			p.Asn866Lys * 16	Frameshift	Uterine Corpus Endometrial Carcinoma	1/530 (0.19%)
4702132	G	A	p.Arg303 *	Stop gained	Uterine Corpus Endometrial Carcinoma	2/530 (0.38%)
4704271	C	A	p.Glu154 * ^1^	Stop gained	Lung adenocarcinoma	1/567 (0.18%)
4695519	C	T	p.Trp404 *	Stop gained	Uterine Corpus Endometrial Carcinoma	1/530 (0.19%)
4685745	C	A	p.Glu638 *	Stop gained	Uterine Corpus Endometrial Carcinoma	1/530 (0.19%)

^1^ Glu154 does not have a corresponding residue in the primary structure of DPP9.

**Table 5 cancers-13-01637-t005:** *DPP9* LoF variants in COSMIC. Nonsense mutation that is a substitution mutation resulting in a premature termination codon (*). CDS = coding sequence; AA = amino acid; SSM = simple somatic mutation.

CDS Mutation	AA Mutation	Legacy Mutation ID	Type	Disease	Number of SSM Affected Cases
c.43G > T	p.Gly15 * ^1^	COSM3960221	Nonsense	Lung squamous cell carcinoma	2
c.180G > A	p.Trp60 *	COSM9150037	Nonsense	Plasma cell myeloma	1
c.306C > G	p.Tyr102 *	COSM4982488	Nonsense	Esophagus squamous cell carcinoma	3
c.460G > T	p.Glu154 * ^1^	COSM7761095	Nonsense	Lung adenocarcinoma	1
c.864C > A	p.Cys288 *	COSM8582330	Nonsense	Stomach carcinoma	1
c.907C > T	p.Arg303 *	COSM998442	Nonsense	Endometrioid carcinoma Colon adenocarcinoma	2
c.976G > T	p.Glu326 *	COSM7365414	Nonsense	Lung adenocarcinoma	1
c.1090G > T	p.Glu364 *	COSM8089586	Nonsense	Malignant melanoma	1
c.1385C > G	p.Ser462 * ^1^	COSM7365412	Nonsense	Lung adenocarcinoma	1
c.2584C > T	p.Gln862 *	COSM7322837	Nonsense	Biliary tract adenocarcinoma	1
c.2668G > T	p.Glu890 *	COSM6866899	Nonsense	Skin squamous cell carcinoma	1

^1^ Gly15, Glu154 and Ser462: do not have corresponding residue in the primary structure of DPP9.

**Table 6 cancers-13-01637-t006:** Genetic variation tolerance in the DPP4 gene family. The o:e ratio, pLI and haploinsufficiency score were retrieved from gnomAD and DECIPHER. Tolerance percentile and GDI values were retrieved from the GeneCards human gene database. o:e, the observed to expected ratio. pLI, the probability of LoF intolerance. GDI, gene damage index score.

Gene	o:e Ratio of LoF	pLI	Haploinsufficiency Score	Tolerance Percentile	GDI
*DPP9*	0.17 (0.1–0.3)	0.99	74.03	7.37%	3.14
*DPP8*	0.14 (0.08–0.27)	1	24.90	26.2%	1.79
*DPP4*	0.44 (0.31–0.62)	0	8.9	15%	2.81
*FAP*	0.86 (0.68–1.1)	0	8.57	10.2%	2.77

## Data Availability

TCGA data are available through Oncolnc and UALCAN (R).

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
