# Peer review of "DPP9: Comprehensive In Silico Analyses of Loss of Function Gene Variants and Associated Gene Expression Signatures in Human Hepatocellular Carcinoma"

_cancers, 2021, doi:10.3390/cancers13071637_

Round 1
Reviewer 1 Report
My concerns have been addressed.
Reviewer 2 Report
The revised manuscript has been improved.
This manuscript is a resubmission of an earlier submission. The following is a list of the peer review reports and author responses from that submission.
Round 1
Reviewer 1 Report
This study examined the distribution of DPP9 variants/mutations in major databases, and tested the association between DPP9 expression and survival. Below is a list of points to be addressed.
- The title said “associations between DPP9 expression and survival” was investigated in this study. And the first sentence of the Discussion said DPP9 variants were investigated for their associations with survival. But no tests for the associations between variant and expression, or between variants and survival, were performed. What the manuscript did was first to summarize variants/mutations provided by public database, and then to examine survival associations of expressions.
- Again, you summarized DPP9 variants/mutations, then what’s the conclusions or implications about those variants/mutations? Are they associated with survivals? What’s your evidence?
- When the title is about hepatocellular carcinoma, then what’s the point of including COVID-19 related variants?
- How did you define “high” co-expression in section 3.6 and 3.7?
- In section 3.6, you said DPP9, DPP8 and DPP4 were stratified based on median expression, then what’s “DPP9,8,4 Low” or “DPP9,8,4 High” in Fig 4? Does “DPP9,8,4 High” mean samples with higher-than-median expressions for all the 3 genes? If yes, so samples with inconsistent expressions, for example “high DPP9, low DPP8, low DPP4”, were not included in the figure?
- Please label sample sizes for each group in survival curves.
- Using Log-rank tests, you claimed that DPP9 expressions were associated with survivals. But DPP9 expressions were also associated with gender, age, tumor stage to some extent respectively. You may need to use Cox proportional hazards model with those clinical factors as covariates to re-evaluate the associations between DPP9 expression and survivals.
- Why did you first performed pathway enrichment analysis using positively correlated genes in common between DPP9 and DPP8, and then selected genes in common amongst DPP9, DPP8 and DPP4? Why did you include two versions for the pathway enrichment analysis?
Author Response
Reviewer #1:
This study examined the distribution of DPP9 variants/mutations in major databases and tested the association between DPP9 expression and survival. Below is a list of points to be addressed.
Major
1 The title said “associations between DPP9 expression and survival” was investigated in this study. And the first sentence of the Discussion said DPP9 variants were investigated for their associations with survival. But no tests for the associations between variant and expression, or between variants and survival, were performed. What the manuscript did was first to summarize variants/mutations provided by public database, and then to examine survival associations of expressions.
Response: We have modified the title and the first sentence of the Discussion to add clarity. The title is changed to
“DPP9: Comprehensive in silico analyses of loss of function gene variants and associated gene expression signatures in human hepatocellular carcinoma”.
In the discussion, on Page 14 Line 363, we changed the sentence to “This is the first set of analyses of human genomic databases to examine DPP9 exonic LoF variants and of associations of DPP9 expression with cancers”.
- Again, you summarized DPP9 variants/mutations, then what’s the conclusions or implications about those variants/mutations? Are they associated with survivals? What’s your evidence?
Response: One of the main discoveries of this study is that we identified DPP9 variants and their associated diseases in three human genetic databases, which are summarised in Tables 3-5. Due to limited case numbers of DPP9 exonic LoF variants, we are unable to associate these DPP9 variants with patient survival. We have modified our manuscript on Page 15 Line 376 to add clarity.
3 When the title is about hepatocellular carcinoma, then what’s the point of including COVID-19 related variants?
Response: The emphasis and majority of this research is HCC, so HCC is in the title. However, the overall research was to comprehensively examine existing databases to increase our understanding of DPP9 in humans from existing data. The DPP9 variants that have been associated with Covid-19 have attracted intense interest, so we believe that in a paper on DPP9 gene variants the readers will want to know more about these prominent variants. In particular, whether there is any existing data on whether these variants have an association with liver disease or HCC. Therefore, these analyses are included.
4 How did you define “high” co-expression in section 3.6 and 3.7?
Response: High co-expression means higher-than-median co-expression, as the high and low mRNA co-expression levels of genes were stratified based on median expression. This is stated in the Methods section (Page 4 Line 142) as well as in section 3.6 and 3.7 as “stratified the expression based on median expression”. Responses to further reviewer questions provide greater clarity.
4 In section 3.6, you said DPP9, DPP8 and DPP4 were stratified based on median expression, then what’s “DPP9,8,4 Low” or “DPP9,8,4 High” in Fig 4? Does “DPP9,8,4 High” mean samples with higher-than-median expressions for all the 3 genes? If yes, so samples with inconsistent expressions, for example “high DPP9, low DPP8, low DPP4”, were not included in the figure?
Response: Yes. “DPP9,8,4 High” mean samples with higher-than-median expressions for all the three genes. Samples with inconsistent expressions, for example “high DPP9, low DPP8, low DPP4”, were not included in the figure. We have modified the method section to clarify it. Please see the changes on Page 4 Line 147.
6 Please label sample sizes for each group in survival curves.
Response: This advice has been accepted and the manuscript has been modified accordingly in the figure legends.
7 Using Log-rank tests, you claimed that DPP9 expressions were associated with survivals. But DPP9 expressions were also associated with gender, age, tumor stage to some extent respectively. You may need to use Cox proportional hazards model with those clinical factors as covariates to re-evaluate the associations between DPP9 expression and survivals.
Response: Based on Figure 2E-G, DPP9 was only significantly associated with gender. As suggested, we used Cox proportional hazards model with gender and BMI as covariates to evaluate associations between DPP9 expression and survival (please see the new Supplementary Tables 5 and 6). It suggests that DPP9 expression was not significantly associated with poor prognosis based on patents’ gender or BMI.
8 Why did you first performed pathway enrichment analysis using positively correlated genes in common between DPP9 and DPP8, and then selected genes in common amongst DPP9, DPP8 and DPP4? Why did you include two versions for the pathway enrichment analysis?
Response: We have investigated 3 sets of positively correlated genes in common among DPP8/9, DPP4/8/9, DPP4/8/9 & FAP, respectively. We decided to perform analyses of DPPs on its own and as a gene family. DPP8 and DPP9 have very similar protein structure, substrate repertoire and biological functions, thus were analysed as a group (Figure 5). DPP4 and FAP are about 50% sequence identical to each other and 27% identical to DPP8 and DPP9. We then added DPP4 into the analysis (Figure 6). Then all four genes were analysed as a family and there were only 7 genes in common (Supplementary Figure 4).
Figure 5B has been modified to be the same version as Figure 6C.
Reviewer 2 Report
This study performed comprehensive in silico analysis to explore the impact of DPP4 gene family, notably DPP9 on HCC.
The authors also explored the impact of age, gender and obesity on DPP9 expression.
And the authors mentioned that the high co-expression of DPP4, 8, 9 and FAP was associated with poor survival in HCC patients.
There are several papers to demonstrate the impact of DPP4 on HCC but the functional roles of DPP9 in HCC are uncertain.
The authors need to add clear explanation and the data to reveal the functional roles of DPP9 on tumor development, age, gender and obesity.
The authors need to add more clear data to make following conclusion: DPP9 is essential for human survival and the DPP4 protease family is important in cancer pathogenesis.
Author Response
- The authors need to add clear explanation and the data to reveal the functional roles of DPP9 on tumour development, age, gender and obesity.
Response: To add clarity, we performed Cox proportional hazards model with gender and obesity (based on BMI category) as covariates to evaluate associations between DPP9 expression and survival. Please see the new Supplementary Tables 5 and 6. DPP9 expression was not significantly associated with poor prognosis based on patents’ gender or patents’ BMI. High DPP9 expression in the overweight group had a hazard ratio of 1.54, but it lacked statistical significance. The manuscript has now been modified in the result and discussion sections, on Page 9-10 and 16.
- The authors need to add more clear data to make following conclusion: DPP9 is essential for human survival and the DPP4 protease family is important in cancer pathogenesis.
Response: We have now modified our conclusion on Page 1, 15 and 16, where highlighted on the tracked document, to support these statements that summarise a large literature.
Reviewer 3 Report
A very nice and interesting analysis. Good utilization of the online databases. In the intro (line 44) be more specific on DPP's role in cancer. For example, in line 52, expand and explain how decreases cell adhesion results in decreased migration. A figure of the DPP4 family - comparing its gene and/or protein structure - would be helpful. Add more explanation to the methods as to how you annotated and compared sequences - is it correct that some databases allowed you to correlate mutations with diseases (TCGA, for example) and others (gnomAD) did not? A figure linking mutation, location, gene, and resulting phenotype or disease would help create a complete picture and connect across the various databases.
Author Response
- In the intro (line 44) be more specific on DPP's role in cancer. For example, in line 52, expand and explain how decreases cell adhesion results in decreased migration.
Response: Introduction has now been modified to elaborate the roles of DPP9 in cancer. Line 44 to Line 52 have now been modified to explain the role of DPPs in cancer.
- A figure of the DPP4 family - comparing its gene and/or protein structure - would be helpful.
Response: A figure has been added as Supplementary Figure 1 to compare protein structure and sequence.
- Add more explanation to the methods as to how you annotated and compared sequences - is it correct that some databases allowed you to correlate mutations with diseases (TCGA, for example) and others (gnomAD) did not?
Response: Method section has now been modified on Page 3 to add more explanation.
- A figure linking mutation, location, gene, and resulting phenotype or disease would help create a complete picture and connect across the various databases.
Response: A figure has been added as Supplementary Figure 2.